# Genome-Wide DNA Methylation Signatures Predict the Early Asymptomatic Doxorubicin-Induced Cardiotoxicity in Breast Cancer

**DOI:** 10.3390/cancers13246291

**Published:** 2021-12-15

**Authors:** Michael A. Bauer, Valentina K. Todorova, Annjanette Stone, Weleetka Carter, Matthew D. Plotkin, Ping-Ching Hsu, Jeanne Y. Wei, Joseph L. Su, Issam Makhoul

**Affiliations:** 1Department of Biomedical Informatics, University of Arkansas for Medical Sciences, 4301 W. Markham St., Little Rock, AR 72205, USA; 2Division of Medical Oncology, University of Arkansas for Medical Sciences, 4301 W. Markham St., Little Rock, AR 72205, USA; MakhoulIssam@uams.edu; 3Pharmacogenomics Analysis Laboratory, Central Arkansas Veterans Healthcare System, 4300 West 7th St., Little Rock, AR 72205, USA; Annjanette.stone@va.gov (A.S.); Weleetka.Carter@va.gov (W.C.); 4Division of Nephrology, University of Arkansas for Medical Sciences, 4301 W. Markham St., Little Rock, AR 72205, USA; mplotkin@uams.edu; 5College of Public Health, University of Arkansas for Medical Sciences, 4301 W. Markham St., Little Rock, AR 72205, USA; PHsu@uams.edu (P.-C.H.); LJSu@uams.edu (J.L.S.); 6Department of Geriatrics, University of Arkansas for Medical Sciences, 4301 W. Markham St., Little Rock, AR 72205, USA; WeiJeanne@uams.edu

**Keywords:** cardiotoxicity, methylation, breast cancer, doxorubicin, cardiomyopathy

## Abstract

**Simple Summary:**

This study examined whether the DNA methylation state of peripheral blood mononuclear cells (PBMCs) could predict cardiotoxicity caused by doxorubicin (DOX)-based chemotherapy in breast cancer patients. The results showed a significant difference in the pattern of DNA methylation of PBMCs associated with a risk of cardiotoxicity. These preliminary findings have the potential to further the goal of personalized medicine and tailor the treatment of breast cancer with DOX-based chemotherapy to reduce the toxicity to the heart.

**Abstract:**

Chemotherapy with doxorubicin (DOX) may cause unpredictable cardiotoxicity. This study aimed to determine whether the methylation signature of peripheral blood mononuclear cells (PBMCs) prior to and after the first cycle of DOX-based chemotherapy could predict the risk of cardiotoxicity in breast cancer patients. Cardiotoxicity was defined as a decrease in left ventricular ejection fraction (LVEF) by >10%. DNA methylation of PBMCs from 9 patients with abnormal LVEF and 10 patients with normal LVEF were examined using Infinium HumanMethylation450 BeadChip. We have identified 14,883 differentially methylated CpGs at baseline and 18,718 CpGs after the first cycle of chemotherapy, which significantly correlated with LVEF status. Significant differentially methylated regions (DMRs) were found in the promoter and the gene body of SLFN12, IRF6 and RNF39 in patients with abnormal LVEF. The pathway analysis found enrichment for regulation of transcription, mRNA splicing, pathways in cancer and ErbB2/4 signaling. The preliminary results from this study showed that the DNA methylation profile of PBMCs may predict the risk of DOX-induced cardiotoxicity prior to chemotherapy. Further studies with larger cohorts of patients are needed to confirm these findings.

## 1. Introduction

Doxorubicin (DOX) is an anthracycline antibiotic commonly used for treatment of breast and other malignancies [1]. A major and yet unpredictable side effect of DOX is the development of cardiomyopathy that may lead to irreversible congestive heart failure [2]. DOX-induced cardiotoxicity is exponentially dose-dependent beginning with the first dose with asymptomatic myocardial injury and may progress to irreversible symptomatic heart failure (HF) years after treatment [3,4]. Incidence of heart failure in cancer patients treated with various DOX-containing regimens has been estimated at 3–45% [5,6]. In total, 10% of patients treated with DOX develop cardiac complications up to 10 years after cessation of chemotherapy [7]. The damaging effects of DOX on the heart often are not detected until years after cessation of the chemotherapy, therefore it is important to identify patients at risk before or at the early doses of chemotherapy [8,9,10].

Because the mechanism of DOX-induced cardiotoxicity is not completely understood, there are no tools to predict or prevent it. The formation of free radicals with oxidative stress is considered a primary mechanism of DOX-associated cardiotoxicity [11], but other mechanisms such as binding to topoisomerases, dysregulation of Ca^2+^ homeostasis, activation of the ubiquitin-proteasome system, release of vasoactive amines, and impaired cardiac repair have also been suggested [12]. Recent studies have focused on DOX-induced systemic inflammation and endothelial injury, which can potentially trigger the development and progression of cardiomyopathy [13,14,15]. We have previously demonstrated that subclinical DOX-induced cardiotoxicity in breast cancer patients was associated with upregulation of transcripts [16] and proteins [17] implicated in immune trafficking and inflammatory response. Further, elevated plasma markers of inflammation, hypercoagulability and endothelial function at baseline and after the first dose of DOX chemotherapy were able seen to be associated with early subclinical DOX-induced cardiotoxicity in patients with breast cancer [18].

Gene expression analysis gives a snapshot at the current state of state of a cell or sets of cells and provide clues to DOX-induced cardiotoxicity. DNA methylation, a major mechanism of gene expression regulation, is a pre-transcriptional heritable modification characterized by the addition of a methyl group to the C5 position of cytosine to form 5-methylcytosine (5 mC) [19]. In somatic cells, DNA methylation occurs predominantly on cytosine residues of the dinucleotide CpG sequences [20]. CpG dinucleotides are distributed unevenly throughout the genome, and in normal, healthy cells 60–80% of CpGs are methylated [21]. Such methylation regulates the stability of the gene expression states and maintains genome integrity by collaborating with proteins involved in gene repression or by inhibiting the binding of transcription factor(s) to DNA [22]. In mammalian cells cytosine is modified to 5 mC by DNA methyltransferases (DNMTs) and TET dioxygenases are responsible for demethylation of 5 mC [23]. A number of studies have shown that changes in the DNA methylation status contributes to biological processes associated with various diseases, such as cardiovascular diseases and cancer [24]. The relationship between DNA methylation and various cardiovascular diseases, including myocardial infarction, acute coronary syndrome, and atherosclerosis has been demonstrated in a series of epigenome-wide association studies [23,25,26].

DNA methylation can affect the pharmacodynamics of various drugs by modulating the expression of specific drug metabolizing enzymes [27]. It has been suggested that DOX might affect the global DNA methylation via dysregulation of mitochondrial function, which is considered a major mechanism of DOX-induced cardiotoxicity [28]. For example, Ferreira et al. [29] showed that treatment of rat H9c2 cardio-myoblast cell line with low sub-therapeutic doses of DOX (10 and 25 nM) caused downregulation of *DNMT1* and global methylation levels, and was associated with cell cycle arrest in G2/M and upregulation of several mitochondrial DNA transcripts. Chronic DOX treatment of rats induced a decrease in the global DNA methylation in the hearts and altered transcript levels of multiple mitochondrial genes, along with increased activity of histone deacetylases [30]. Nordgren et al. [31] showed that chronic DOX exposure of rats altered DNA methylation of 14 genes in the heart tissue, of which 5 genes (*RBM20*, *NMNAT2*, *KLHL29*, *CACNA1C*, *SCN5A*) were significantly altered in the gene expression level, including down-regulation of *KLHL29*, *NMNAT2*, and *SCN5A*, and up-regulation of *RAB20* and *CACNA1C*. Hoefer et al. [25] found that DNA methylation of DOX-metabolizing enzyme AKR7A2 in hearts of patients treated with DOX, impacted the expression of AKR7A2 and the synthesis of cardiotoxic DOX metabolite daunorubicinol [25]. DNA methylation may play a key role in cardiac function impacted by DOX-based chemotherapy.

This preliminary study aimed to examine whether DNA methylation of PBMCs could predict cardiotoxicity of DOX-based chemotherapy in breast cancer patients.

## 2. Materials and Methods

### 2.1. Study Subjects and Blood Samples

Patients with early breast cancer eligible for DOX-based chemotherapy were enrolled at the Winthrop Rockefeller Cancer Institute, UAMS. This study was approved by the Institutional Review Board (IRB) of UAMS (Protocol #130212) and from IRB of the Central Veterans Healthcare system (CAVHS) (Protocol #1423976-2), where the samples were processed and stored. The study was performed on a total of 19 subjects enrolled in the study between 2012 and 2014. All participants signed an IRB approved informed consent where they were informed for the use of their blood samples and medical records for research purposes. The inclusion criteria included early ER+/PR+/Her2−, ER+/PR−/Her2− or triple negative, stage I to III breast cancers within 18–99 years of age. Participants were ineligible if they were pregnant or breast feeding and had prior history of chemotherapy or radiotherapy. All patients were treated with a predefined protocol which included a combination of DOX (60 mg/m^2^) with cyclophosphamide (600 mg/m^2^) in each cycle for 4 cycles every 2 weeks. Patients with hypertension who were taking antihypertensive medications (β-blockers and ACE inhibitors) prior to chemotherapy were prescribed to continue with this treatment concomitant with the DOX-based chemotherapy. Patients with diabetes also continued to be treated with insulin or metformin concomitant with the chemotherapy.

Blood samples were collected prior to chemotherapy and after the first cycle of chemotherapy. PBMCs were isolated from EDTA anti-coagulated blood using standard Ficoll-Paque Plus gradient centrifugation (density 1.073 g/mL) according to the instructions of the manufacturer (GE Healthcare, USA), and as described in the application note [32]. Briefly, EDTA anti-coagulated blood, diluted with an equal volume of phosphate-buffered saline (PBS) was layered over the Ficoll-Paque Plus and was centrifuged at 400 g for 30 min at 18 °C–20 °C with the brake off. After removing the upper layer containing plasma and platelets, the layer of PBMCs was isolated and stored at −80 °C until further use.

### 2.2. Assessment of Left Ventrical Ejection Fraction as a Measure of Cardiac Function

Cardiac toxicity was evaluated by clinical assessment of LVEF with MUGA scan before and after the fourth cycle of DOX-based chemotherapy. A decline of LVEF by >10% or below 50% in comparison with the baseline (before the start of chemotherapy) was considered abnormal [33,34].

### 2.3. DNA Extraction and Methylation Measurements

DNA was extracted from PBMCs using the QIAamp DNA Blood Mini Kit (Qiagen, Valencia, CA, USA), following manufacturer’s instructions. Samples were quality assessed and quantified by ultraviolet (UV) absorbance measured via Nanodrop Technologies, NanoDrop^®^ ND-2000 Spectrophotometer (Wilmington, DE, USA) and the software. The integrity and quantity of the DNA samples were determined by TaqMan^®^ RNase P Detection assay (Applied Biosystems Assay, Life Technologies, Carlsbad, CA, USA) with fluorescence detection on a 7900 Fast Real Time PCR System (Applied Biosystems, Life Technologies, Carlsbad, CA, USA) per the manufacturer’s protocol.

For genome-wide analysis of DNA methylation, samples were bisulfite-modified using Zymo EZ-96 DNA Methylation Kits (D5004). The bisulfite-mediated conversion efficiency was determined by PCR with DAPK1 primers (Zymo) and gel electrophoresis of PCR-products [35]. The bisulfite-modified DNA samples were whole-genome amplified, fragmented, precipitated, resuspended, and hybridized to Illumina Human Methylation 450 beadchips, which simultaneously profiles the methylation status for >485,000 CpG sites at single-nucleotide resolution and covers 96% of CpG islands with additional coverage of island shores (<2 Kb from CpG Islands), island shelves (2–4 Kb from CpG islands), and regions flanking them. The beadchips were scanned using the Illumina iScan System.

### 2.4. Data Quality Control and Normalization Pipeline

The resulting raw intensity data, as IDAT files (the Illumina proprietary file format used to store data output directly from the scanner), were imported into the ChAMP R package [36], which was used for the processing and analysis of the methylation arrays using default values. The Beta-Mixture Quantile (BMIQ) [37] normalization method was used to normalize the array data. This method is an intra-array normalization strategy. Batch effects were identified and corrected using the COMBAT algorithm. Probes on a blacklist of probes that are known to be cross-reactive were removed.

### 2.5. Methylation Data Analysis

Two different methylation analysis tools, CpGAssoc [38] and ChAMP were used to identify significantly differentially methylated probes. We also used the ChAMP software package functions for differential methylation probe (DMP) and differentially methylated region (DMR) analysis. The DMP analysis algorithm internally uses linear models for microarray data (LIMMA) [39]. The DMR analysis function uses the Bumphunter [40] R package. The R package CpGAssoc [38] was used to identify probes significantly associated with ejection fraction (LVEF) status. CpGAssoc uses a mixed or fixed effect model to identify probes that are significantly associated with cardiotoxicity. Both analyses used normalized and batch corrected beta values as input. Figure 1 shows an overview of the analysis performed.

### 2.6. Gene Expression Validation by Real-Time Quantitative PCR (QPCR)

Real-time QPCR was performed as described by Plotkin et al. [41]. Total RNA was isolated from PBMCs of 10 breast cancer patients, including 5 patients with normal LVEF and 5 patients with abnormal LVEF before, and after the first cycle of DOX-based chemotherapy using RNeasy mini kit (Qiagen, Valencia, CA, USA), following manufacturer’s instructions. gDNA was removed with gDNA eliminator columns. Concentrations (ng/μL) and OD ratios (260/280 nm) of total RNA were determined using the Nanodrop UV/VIS spectrophotometer (Thermo Fisher). RNA integrity number scores, which are a ratio of ribosomal RNAs 18S and 28S in total RNA samples, were obtained using the Agilent 2100 Bioanalyzer with the Agilent RNA 6000 Nano Kit (Santa Clara, CA, USA). All total RNA specimens had OD ratios of 1.85 to 2.10 and RNA integrity number scores of >7.4. The quantitative conversion of 250 ng of total RNA to single-stranded cDNA in a single 30-μL reaction was performed with the High Capacity cDNA Reverse Transcription Kit (Applied Biosystems, Foster City, CA, USA). Quantitative PCR was performed using QuantStudio 12K Flex real-time PCR system software version 1.3 (Applied Biosystems). TaqMan gene expression assays were Hs01062178_m1 (IRF6); Hs00430118_m1 (SLFN12); Hs00961882_m1 (RNF39); Hs99999903_m1 (beta-actin) and Hs00976258_m1 [general transcription factor 2B (GTF2B)]. All quantitative PCRs were performed in a final volume of 10 μL containing 1× of TaqMan Gene Expression Master mix with UNG (Applied Biosystems), 1× of each TaqMan Gene Expression Assay (FAM-MGB dyes), and 10 ng cDNA in sterile molecular-grade water. The standard cycling conditions were 50 °C for 2 min, 95 °C for 10 min, followed by 40 cycles of 95 °C for 15 s, and 60 °C for 1 min. Quantitative PCR was performed in triplicate to ensure quantitative accuracy. The results were analyzed using Expression Suite Software version 1.0 (Applied Biosystems). Relative expression levels were calculated for each sample after normalization against the housekeeping genes beta-actin and GTF2B, using the ΔΔCt method for comparing relative fold expression differences. The data are expressed as fold change (FC).

### 2.7. Pathway and Gene Ontology Analysis

Functional annotation and enrichment analyses were conducted using Ingenuity Pathways Analysis (IPA) software (Ingenuity Systems; www.ingenuity.com/ (accessed on 9 March 2021)). WEBGestalt [42] was used for gene set enrichment analysis (GSEA). Additionally STRING [43] and NetworkAnalyst [44] were used to build protein-protein networks and gene regulatory networks to explore the significant probes and associated genes.

### 2.8. Statistical Analysis

Fisher’s exact test was used to test for differences between the patient groups. A *t*-test was performed to determine differences in LVEF change between the groups. In the methylation analysis, we account for multiple testing using the Holm-Bonferroni method or false discovery rate (FDR) and results were considered significant with adjusted *p*-values or FDR ≤ 0.05. Beta values were used to describe the amount of methylation with a range from 0 (no methylation) to 1 (complete methylation).

## 3. Results

### 3.1. Demographic Characteristics of the Study Participants

The characteristics of the patients are presented in Table 1. Of the 19 patients enrolled, 9 patients had asymptomatic LVEF > 10% decrease in comparison with the baseline (abnormal group) and 10 patients had LVEF ≤ 10% (normal group). The median change of LVEF among the abnormal group was 14%, while in the normal group the median change in LVEF was 0.6%. A two-tailed *t*-test of the delta in LVEF before and after treatment between the two groups was significant with a *p*-value ≤ 0.05 (Appendix A). In the abnormal group, 3 patients had hypertension, versus 4 with hypertension and 1 with diabetes in the normal group. 

No significant differences were detected between the two groups of patients (normal and abnormal) with respect to the age, race, and type of breast cancer. 

### 3.2. Differential Methylation Analysis Identified Probes Associated with Abnormal Ejection Fraction

To initially investigate the quality of the batch correction and normalization and to identify any confounding factors we performed principal component analysis (PCA). We took the top 10,000 most variable methylation sites across all samples as determined by the probes that showed the highest variance. There was no obvious batch effect or confounding factors, such as race, but we did observe a clear separation between the normal and abnormal LVEF samples, Figure 2A. There was no separation of sample at baseline and after the first cycle of DOX. Taking the average beta values of probes that had a beta delta of 20% or greater we see clear methylation differences between the two groups, Figure 2B.

We looked at baseline and after the 1st cycle of DOX samples separately comparing the normal versus the abnormal LVEF samples. Using the ChAMPDMP R tool, we identified 15,483 significant probes from baseline (normal vs. abnormal), and 20,736 significant probes from the comparison after the 1st cycle of DOX (normal vs. abnormal), both results with an adjusted *p*-value ≤ 0.05. To verify these results, we used a second algorithm to identify significant probes associated with LVEF status. We identified 14,883 differentially methylated CpGs at baseline (Figure 3A) and 18,718 CpGs after the first cycle (Figure 3B) which were significantly correlated with LVEF status (adjusted *p*-value ≤ 0.05). 

Downstream analysis involved these significantly differentially methylated probes found between the normal and abnormal LVEF groups. The significant probes were uniformly distributed across all autosomal chromosomes (sex chromosomes were excluded). Intersecting the 4 results resulted in 5139 differentially methylated probes, (Appendix A). The 5139 probes were annotated close to 3905 genes (2978 unique genes). Figure 4A shows the genomic feature distribution of the significant DMPs with 33% of the probes associated with the gene body feature. The top 5 significantly differentially methylated probes annotated with genes included RGS14 (3 probes), KLH31, and ANO4 (Figure 4B).

Using the online tool WEBGestalt, we performed GSEA using the 2978 significant genes and the beta change between the normal and abnormal LVEF as input and identified several enriched pathways. The two most significantly enriched pathways, after multiple testing correction with positive enrichment score involved mRNA splicing, which included the following genes: CACS3, DHX15, DHX9, EIF4A3, HNRNPH1; HNRNPR; HNRNPU; RBM17; SF1; SNRPN, and YBX1, and the most significant negatively enrichment score was Interferon gamma signaling (IFN-γ), which included HLA-DRB1, TRIM14, HLA-A, HLA-F, IRF6 (Figure 5).

### 3.3. Top 100 Significant Probes Cluster Samples Based on Ejection Fraction

Unsupervised hierarchical clustering of the top 100 significant differentially methylated probes with a delta beta of at least 10% shows a clustering of the samples into two groups based on LVEF status (Figure 6A). Using the online tool NetworkAnalyst, the genes associated with the 100 probes were input as seed genes and mapped to protein–protein interaction networks. The largest subnetwork contained 23 of these top genes highly connected with only 1 degree (one non-seed gene) of separation from each other (Figure 6B). Enrichment analysis was performed on this subnetwork looking at the Gene Ontology, Reactome and KEGG Pathways databases. Among the identified significant enriched pathways, we found regulation of transcription, mRNA splicing, pathways in cancer and signaling by ErbB2/4 (Figure 6C).

### 3.4. Significant Region of Differential Methylation

We next looked for regions of consistent differential methylation between abnormal and normal LVEF. We identified 185 DMRs including highly significant differences where 82 (44%) overlapped the 5′ region of the gene and 36 (19%) where found in the promotor region. Significant DMRs included regions near SLFN12, IRF6, and RNF39. IRF6 and SNFN12, which encodes interferon regulatory factor 6 and SNFN12 (Schlafen Family Member 12), were identified as significantly increased in methylation of the abnormal LVEF group (Figure 7A,B). Whereas the protein coding RNF39 (Ring Finger Protein 39) was significantly less methylated in the abnormal LVEF group (Figure 7C).

### 3.5. Gene Expression of IRF6, SLFN12 and RNF39

We used Taqman qPCR to investigate the correlation between the differential methylation of IRF6, SLFN12, and RNF39 with their gene expression at baseline and after the first cycle of DOX chemotherapy in PBMCs of breast cancer patients (Appendix A). The hypermethylation of *IRF6* and *SLFN12* in patients with abnormal LVEF was associated with downregulation of both *IRF6* and *SLFN12* after the first chemotherapy cycle versus the baseline in all of the examined samples. The mean values of FC with standard error (SE) are shown on Figure 8. 

### 3.6. IPA Pathway Analysis

In order to define the potential relationships between the differentially methylated genes between patients with a risk for cardiotoxicity and patients without, we used the IPA platform. The input for IPA was the 73 genes associated with the top 100 significant DMP with the delta beta value used in place of expression value. IPA analysis found 67 network-eligible genes in 6 networks, related to: cell morphology and organization, post-translational modification, connective tissue disorders, endocrine system disorders. Figure 9A shows the three most significant networks of the differentially methylated genes, “Cell Morphology, Cellular assembly and organization, post-translational modification”; “Connective tissue disorders, developmental disorder, hereditary disorder” and “Endocrine system disorders, organ morphology, organismal functions”. IPA identified 10 overlapping canonical pathways (Figure 9B) which share 2 differentially methylated genes, coding for calcium–activated serine/threonine protein kinases C alpha (PKC-alpha) and PKC-eta. The most significant canonical signaling is the Vitamin D Receptor (VDR)/Retinoid X Receptor (RXR) activation pathway. Three of the top canonical pathways were predicted to be upregulated, including ErbB signaling, Estrogen receptor (ER) signaling, and Thrombin signaling.

## 4. Discussion

In this study, we investigated the potential of methylation state to shed light on predicting cardiotoxicity of DOX-based chemotherapy in breast cancer. The preliminary results showed, for the first time, to our knowledge, that PBMCs’ DNA methylation prior to the start of the treatment could predict the risk of DOX-induced cardiotoxicity in breast cancer patients.

In our analysis significant methylation differences were identified between patients that later developed abnormal LVEF and those that did not, both at baseline and after the first DOX treatment cycle, suggesting that patients with those epigenetic signatures could be predisposed to cardiotoxic side effects. The differences in the methylation profiles were seen across the genome and affected several genes and pathways. Many of the most significantly differentially methylated genes are the ones that have been shown to be involved in heart and cardiovascular diseases. Regulator of G Protein Signaling 14 (*RGS14*) is a protein coding gene that plays a role in the signaling by GPCR and Ras signaling pathway, as well as serving as an inhibitor of platelet-derived growth factor (PDGF)-stimulated ERK1/ERK2 phosphorylation. This gene has been implicated in cardiovascular disease and has been shown to play a role in cardiac remodeling [45,46]. The protein coding gene, Kelch Like Family Member 31 (*KLHL31*) inhibits the transcriptional activities of TPA-response element (TRE) and serum response element (SRE). TRE has been shown to be a transcriptional repressor in mouse cardiomyocytes [47], making it an interesting gene to further investigate for its potential role in DOX induced cardiotoxicity. Anoctamin 4 (*ANO4*) is a protein coding gene involved in ion channel transport. In a study that looked at chronic DOX exposure in rats, it was found to be significantly differentially methylated between the treated versus untreated group [31]. 

The results of GSEA have shown a strong enrichment in transcription regulation and in particular mRNA splicing. Importantly, we observed that the alternative splicing may already be different even before DOX treatment and may thus be a potential reason for the differences in response. Growing evidence has demonstrated that the response differences in several treatments for cancer can be traced back to differences in the alternative splicing, which can result in completely different proteins with opposing functions [48]. The two most significantly affected pathways, after multiple testing corrections, were mRNA splicing and IFN-γ signaling. Alternative splicing, an important mechanism to generate transcriptomic and proteomic diversity from the genome, has emerged as a crucial process governing biological processes during cardiac development and disease progression [49]. Overexpression of *DHX15* [50], *EIF4A3* [50], and *RBM17* [51] splicing factors have been demonstrated in cancer. EIF4A3 [52] and *DHX15* [53] have been implicated in the contractile function of cardiomyocytes. *HNRNPu* [54] has been found to be required for normal pre-mRNA splicing and postnatal heart development, and function. IFN-γ signaling is primarily associated with inflammation and cell-mediated immune response, but also in the promotion of tumor progression [55]. *IRF6*, a member of the IFN family of transcription factors is one of the significantly hypomethylated genes prior to the start of chemotherapy that predicted the risk of DOX-induced cardiotoxicity in our study. This finding correlate with previous reports showing that *IRF6* has a protective role in the response to endotoxic shock [56], which is one of the suggested mechanisms of DOX-induced inflammation and multiorgan toxicity [57]. Downregulation of *IRF6* has been demonstrated in several cancers, suggesting tumor-suppressor functions. Downregulation of IRF6 was demonstrated in highly invasive breast cancer cell lines and when elevated, it suppressed cell proliferation, and enhanced sensitivity to chemotherapy [58], but the impacts of IRF6 on cardiovascular diseases remain largely unknown [59]. This study showed that IRF6 and SLNF12 hypermethylation correlated with increased risk of DOX-induced cardiotoxicity. We also found that the DNA hypermethylation of IRF6 and SLFN12 in patients with abnormal LVEF was associated with reduced gene expression, a finding which correlate with the generally accepted effect of hypermethylation [60]. The expression of RNF39, which is hypomethylated in patients with abnormal LVEF was not detected in any of the samples by qPCR. A possible explanation could be the low expression of RNF39, due to DOX-induced reduction of B-lymphocytes [61], which are the main leukocytes expressing RNF39 [62].

Pathway analysis showed that differentially methylated CpGs of the group of patients with DOX-induced cardiotoxicity were associated overlapping pathways, which included hypermethylated PKC-alpha (PRKCA) and PKC-eta (PRKCH). Several studies have demonstrated the role of PKC isoforms in cardiovascular diseases [63,64]. Activation of PKC-alpha and several other PKC isoforms is regulated during heart hypertrophy and heart failure, making them therapeutic targets for treatment of cardiovascular diseases [65,66]. It has been demonstrated that PKC-alpha physically interacts with and phosphorylates DNMT1, suggesting its possible roles in the control of DNA methylation patterns of the genome, and possibly in the control of gene expression [67]. DNMT1 activity could be regulated at post-translational level through phosphorylation by a serine/threonine kinase, leading to a global hypomethylation in cancer [68]. Both PKC-alpha and PKC-eta have also been associated with poor prognosis and resistance to chemotherapy in breast cancer [69,70,71]. The VDR/RXR pathway has been implicated both in cancer and cardiovascular diseases. The transcriptional factor VDR in a complex with 1,25(OH)_2_D_3_ and RXR regulates the expression of genes involved in cell proliferation and differentiation, oxidative stress, and apoptosis [72], and is expressed in cardiomyocytes [73] and various other cell types [74,75]. Vitamin D deficiency has been associated with a number of cancers, hypertension, coronary artery diseases, diabetes, and autoimmune disorders [76]. ErbB receptor tyrosine kinases epidermal growth factor receptor (EGFR) and ErbB2 (neu, HER2) are often overexpressed, amplified, or mutated in many forms of cancer, including breast cancer, making them important therapeutic targets [77,78]. Overexpression of the erbB-2 in breast tumors has been suggested to be a predictor of the therapeutic response to DOX [79,80,81]. At the same time, erbB2 overexpression in the heart leads to hypertrophy [82]. Treatment of rats with DOX resulted in a dose-dependent increase in ErbB2 in the hearts before the evidence of functional systolic deficit, and was associated with activation of Akt signaling [81]. The crosstalk between ER signaling and other signaling pathways is believed to affect the development of mammary gland and breast tumor initiation and invasion [83,84]. It is well known that ER signaling plays an important role in breast cancer progression and the majority of the human breast cancers start out as estrogen dependent [85]. The mechanism of ER action involves estrogen binding to receptors in the cytoplasm, followed by dimerization of the receptor and translocation to the nucleus, where it binds to estrogen response element (EREs) near the promoters of the target genes, such as *GATA3*, *ATF2*, *AREG*, *GREB1*, *ESRRB*, *FOXA1* [86,87]. The analysis of the significantly methylated genes showed upregulation of thrombin signaling. The activation of the coagulation cascade in which thrombin plays a key role is closely related to inflammation, development of cardiovascular diseases and heart failure prognosis [88]. Cancer patients are at increased risk of thrombosis [89,90], including patients with breast cancer. Furthermore, cancer chemotherapy increases the risk of cancer-related thrombosis, which is a major risk factor for cardiovascular diseases [91,92]. DOX induces severe inflammatory responses in various organs including liver, kidney, intestine, and blood vessels, in addition to its major adverse effect of cardiotoxicity [57]. Accordingly, our previous study demonstrated that elevated markers of inflammation, hypercoagulability and endothelial function (i.e., thrombomodulin, myeloperoxidase, thrombin–anti-thrombin complex) prior to and after the first dose of DOX chemotherapy were able to predict the early subclinical DOX-induced cardiotoxicity in patients with breast cancer [18]. 

The limitations of this preliminary study include the small number of patients examined, which resulted in great variation in the resulting data and a weaker correlation with cardiotoxicity Therefore, indicating the need for further studies with a larger group of patients. In addition, studies that investigate the dynamic profile of the suggested markers of hypercoagulability and endothelial dysfunction during the course of DOX chemotherapy in correlation with the risk of cardiotoxicity are needed.

## 5. Conclusions

Breast cancer is the most common neoplasm in women and the second leading cause of cancer-related mortality in females worldwide [93]. At present, breast cancer detection relies mostly on mammography, which has been associated with decreased breast cancer mortality; however, mammography screening has generated controversy due to the risks of false-positive results and over-diagnosis of indolent disease [94,95]. A population-based study of breast cancer survivors showed that women who received anthracyclines and had more than 10 years of follow-up experienced higher rates of heart failure than did women who received non-anthracycline or no chemotherapy [96]. These observations raise concerns that adult-onset cancer survivors might be plagued by increased cardiovascular morbidity similar to that of long-term survivors of childhood cancer [4]. The need for the means to detect early signs of cardiac deterioration that are related to subsequent clinically significant cardiovascular events is urgent.

The preliminary results from this study provide evidence that the DNA methylation profile of PBMC has the potential to predict the risk of DOX-induced cardiotoxicity. The important finding was that the extent of methylation at baseline correlated with the post-DOX LVEF reduction, indicating that such methylation profiles may have the potential to predict the subsequent development of cardiotoxicity. The number of significant differentially methylated results provides a number of interesting potential markers. Further studies with a larger cohort of patients are needed to confirm these findings, as well as narrowing down candidate methylation markers that can be implemented in a blood test. These finding may help guide treatment or identify patients that need to be followed closely to mitigate heart damage after DOX treatment.

## Figures and Tables

**Figure 1 cancers-13-06291-f001:**
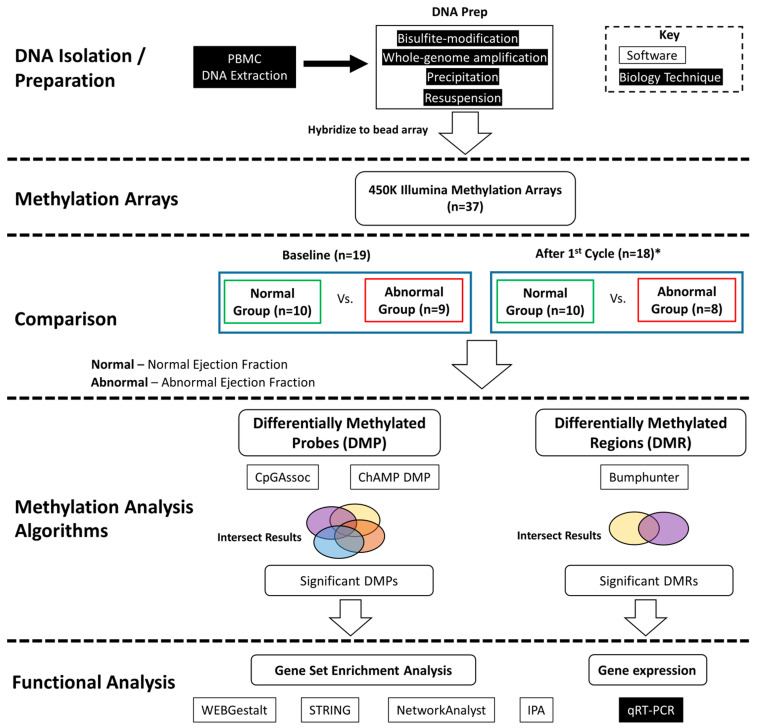
Analysis flow diagram. DNA was extracted from PBMCs and prepared for hybridation to the methylation arrays. Differential methylation analysis was performed between the normal versus abnormal ejection fraction samples at baseline and after first cycle and the intersection of results are reported. Significantly differentially methylated probes (DMPs) and regions (DMRs) were identified. Functional analysis was performed on genes associated with significant differential methylation. * One sample did not have post 1st cycle methylation data.

**Figure 2 cancers-13-06291-f002:**
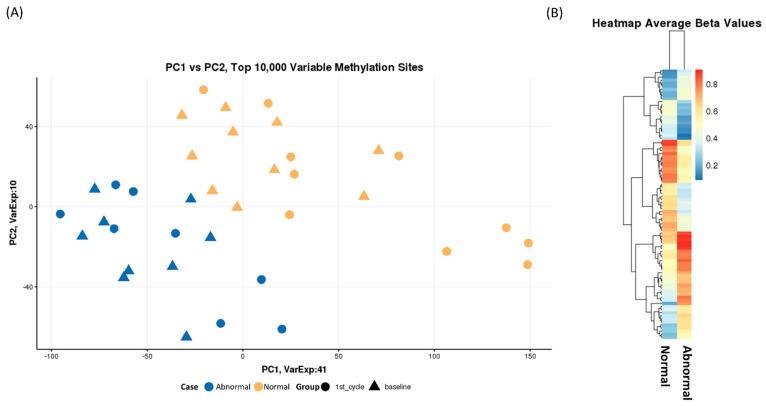
(**A**) Principal Component Analysis of the top 10,000 most variable probes across each sample. Over 50% of variaion is explained by principal component 1 and 2. Samples cluster togther based on case (normal/abnormal ejection fraction). (**B**) A heatmap of average beta values of probes with a difference in beta of greater than 20% between normal and abnormal ejection fraction status. We observe a distinct pattern of methylation between the two groups.

**Figure 3 cancers-13-06291-f003:**
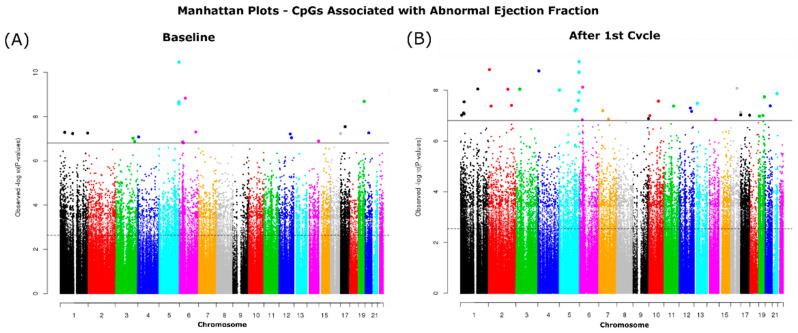
Manhattan plots of CpGs significantly associated with ejection fraction status. Associated CpGs were identified at (**A**) Baseline and (**B**) after 1st cycle. The dotted horizonal line indicated significance by FDR (*p*-value ≤ 0.05) and the solid line indicates significance by the more conservative p adjustment method Holm-Bonferroni (*p*-value ≤ 0.05).

**Figure 4 cancers-13-06291-f004:**
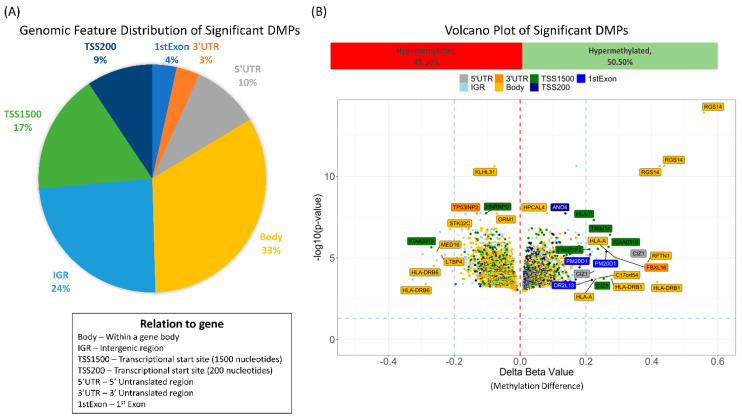
The results from the analysis normal vs. abnormal at baseline and normal vs. abnormal after first cycle using two different algorithms were intersected. This resulted in 5139 in common significantly differentially methylated probes. (**A**) Pie chart showing the genomic feature distribution. (**B**) Volcano plot of the BMA methylation changes (delta beta) in the abnormal ejection fraction versus normal colored by genomic feature. In total, 49.5% of the differentially methylated probes were hypomethylated and 50.5% were hypomethlated as compared to the normal ejection fraction group.

**Figure 5 cancers-13-06291-f005:**
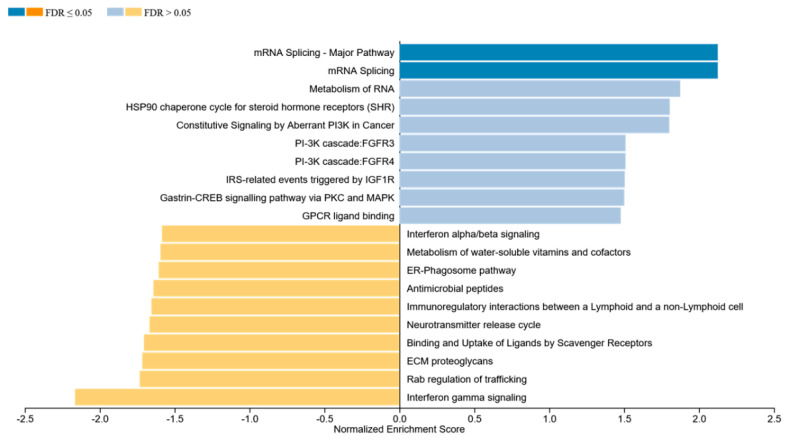
GSEA was performed using the online tool WEBGestalt. The 2978 genes that were associated with significant differentially methylated probes were used as input along with the beta change between the normal and abnormal LVEF to identify enriched pathways.

**Figure 6 cancers-13-06291-f006:**
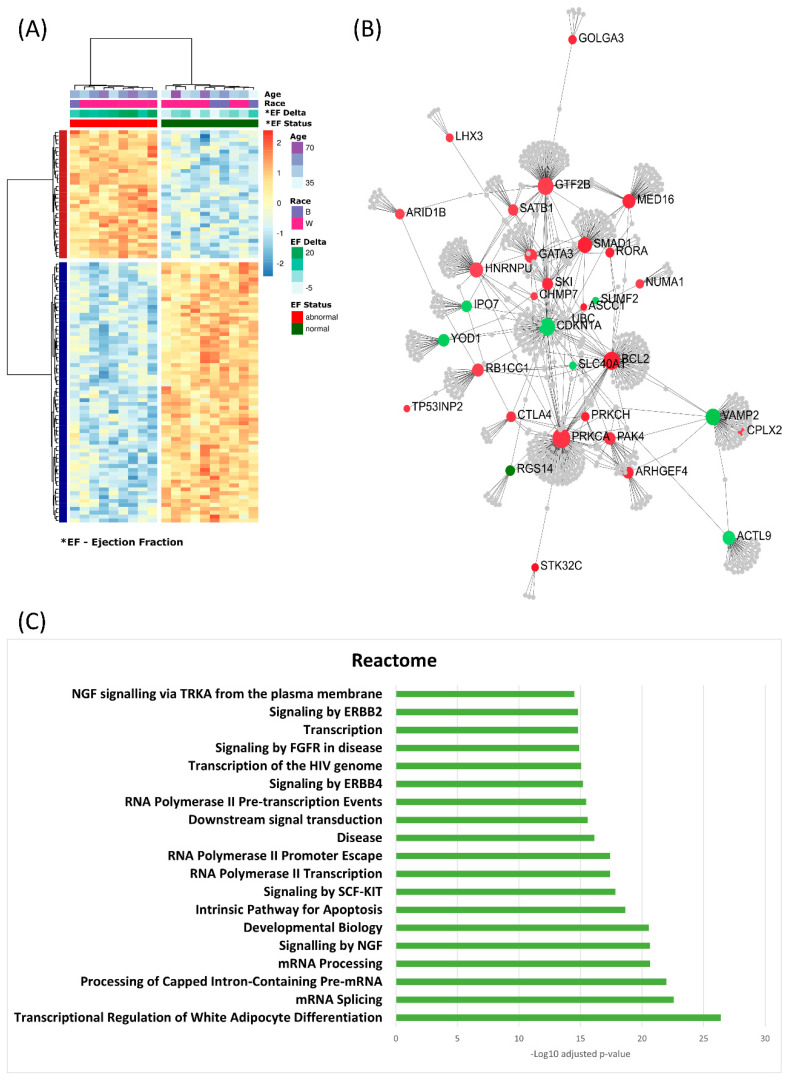
Functional analysis of the top significantly differentially methylated probes. (**A**) Heatmap of the results of unsupervised hierachial clustering using the top 100 probes found by all 4 analyses with at least a 10% beta delta. The clustering reveals differentially methylated probes that distinguish between samples based on ejection fraction status. (**B**) NetworkAnalysis was used on the genes associated with the top probes and mapped to protein–protein interaction (PPI) networks. In total, 23 genes created a network with PPI network 1 degree connection (addition of 1 gene to make the connection). (**C**) Gene Set Enrichment Analysis on the genes in the network was conducted using Reactome pathway database.

**Figure 7 cancers-13-06291-f007:**
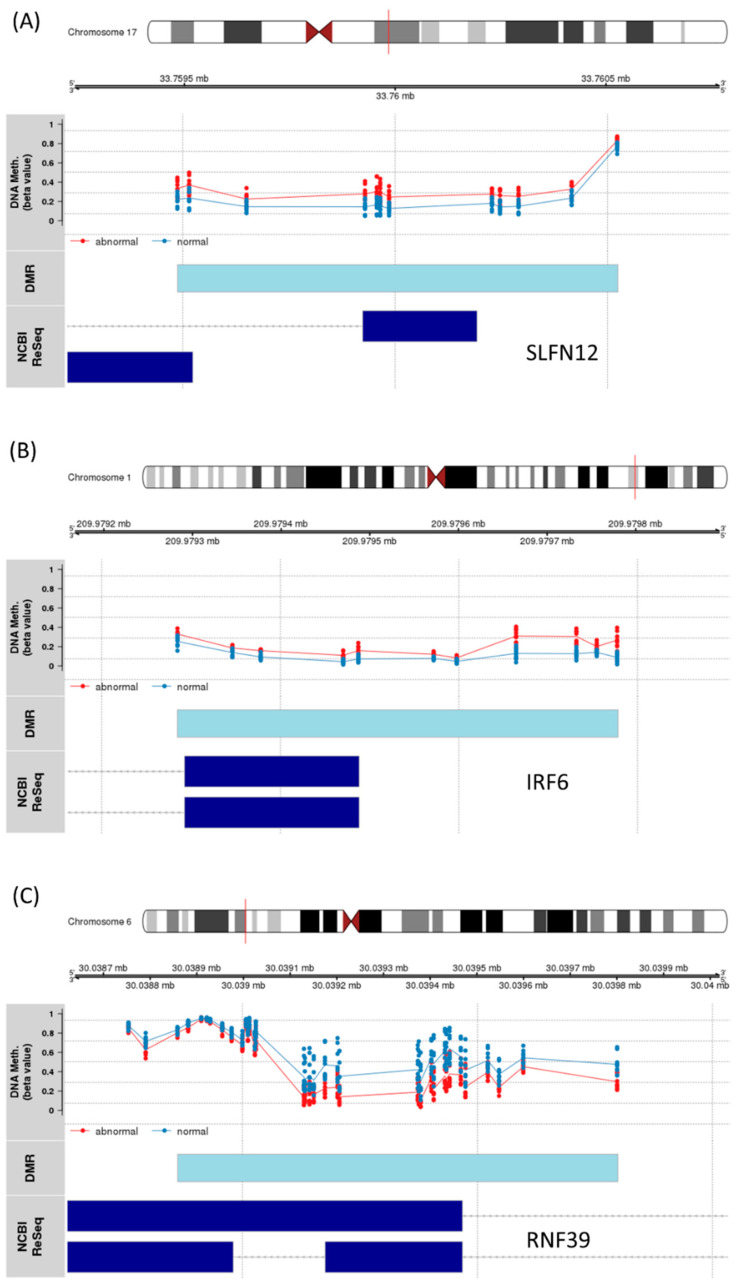
Significant DMR results (**A**) Schlafen Family Member 12 (SLFN12), (**B**) Interferon Regulatory Factor 6 (IRF6), and (**C**) Ring Finger Protein 39 (RNF39). The location of the DMR (light blue) bar. The probes associated with the DMR are plotted by their beta value and colored by patient group ejection fraction status (blue—normal, red—abnormal). The dark blue bars indicate the exon for the different gene transcripts as defined in the RefSeq database.

**Figure 8 cancers-13-06291-f008:**
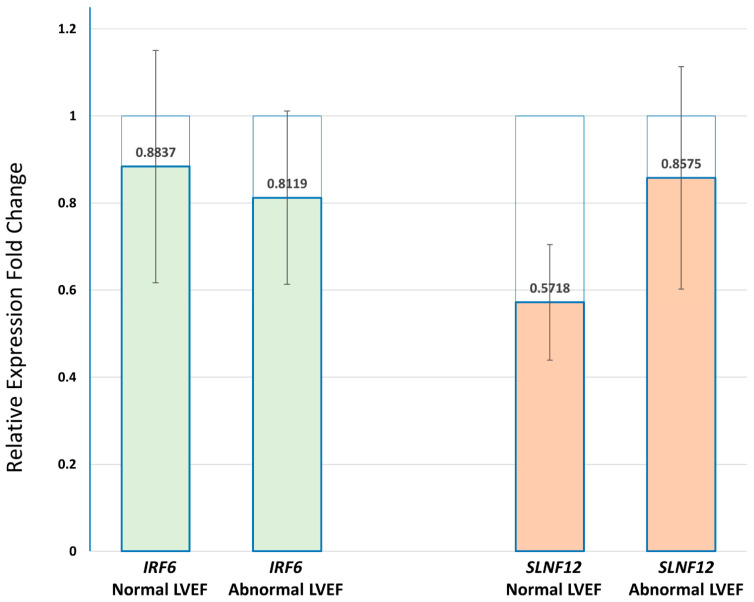
Real-time QPCR of IRF6 and SLFN12 in PBMCs of breast cancer patients with normal and abnormal LVEF following DOX-based chemotherapy. The transcripts of both IRF6 and SLNF12 genes were decreased after the first chemotherapy cycle when compared to before the start of chemotherapy in both groups of patients (*p*-value > 0.05). Relative expression levels were calculated for each sample after normalization against the housekeeping genes beta-actin and GTF28. Experiments have been conducted in triplicates. Similar results were obtained using the housekeeping genes beta-actin and GTF2B.

**Figure 9 cancers-13-06291-f009:**
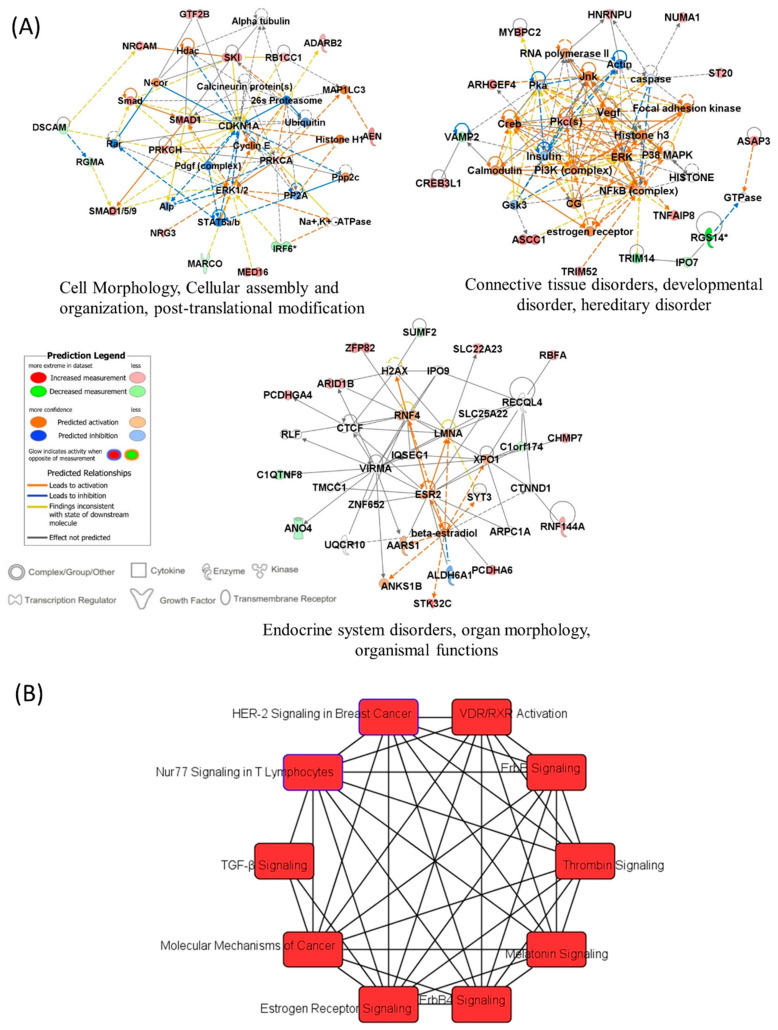
IPA results (**A**) Most significant gene networks of the differentially methylated genes, which visually represent the connections between the genes associated with DOX-cardiotoxicity. The intensity of the node color indicated the degree of up- and down-regulation and uncolored show the genes not relevant to the IPA network database. (**B**) Top ten canonical pathways enriched with the differentially methylated genes.

**Table 1 cancers-13-06291-t001:** Patient Characteristics.

Patient Characteristics			
Characteristics	Normal * LVEF (*n* = 10)	Abnormal * LVEF (*n* = 9)	† Test
Age (average, range)	53.4 (35–73)	50.6 (43–66)	NS (0.234)
Race			
European American	7	8	NS (0.582)
African American	3	1	
Breast Cancer			
ER-/PR−/Her2−	3	1	NS (0.810)
ER+/PR−/Her2−	1	1	
ER+/PR+/Her2−	6	6	
ER+/PR+/Her2+		1	
LVEF Baseline (average %, Std. Dev)	62.8, 6.0	67.7, 7.2	NA
LVEF After 4 cycles (average %, Std. Dev)	62.4, 8.0	53.8, 7.2	NA

* LVEF Left ventricular ejection fraction; † Fisher’s exact test.

## Data Availability

The data discussed in this publication have been deposited in NCBI’s Gene Expression Omnibus and are accessible through GEO Series accession number GSE178887 (https://www.ncbi.nlm.nih.gov/geo/query/acc.cgi?acc=GSE178887 (accessed on 1 October 2021)).

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
