# Peer review of "Genome-Wide DNA Methylation Signatures Predict the Early Asymptomatic Doxorubicin-Induced Cardiotoxicity in Breast Cancer"

_cancers, 2021, doi:10.3390/cancers13246291_

Round 1

Reviewer 1 Report

The chemotherapy with anthracycline in breast cancer patients might induce cardiotoxicity. The authors want to predict the emerging cardiotoxicitic risk by using genome-wide DNA methylation signatures before and after treated with first cycle of doxorubicin. At first, they separated 19 patients into normal and abnormal group by the change of LVEF. They found the DNA methylation signatures correlate with abnormal group. Through some bioinformatics analyses, they conclude the significant differential methylation regions in RNA39, IRF6 and SLFN12 promoter regions. The doxorubicin-induced cardiotoxicity is a serious problem in the therapy of breast cancer patients, it is good if there is a good predicted method to identify the patients with a possible risk. However, I think this manuscript did not achieve this aim.

Major Comments:

  1. The authors should verify the results of DMR by (1) confirmed the differential methylation in patients by bisulfite sequencing or PCR-restriction enzyme digestion in RNA39, IRF6 and SLFN12 promoter region; (2) performing qPCR analysis to monitor their gene expression. These are important to demonstrate the relationship between DMR and cardiotoxicity.
  2. The aim 1 and aim 2 descriptions are confused. I suggest that just mention to identify the DMRs in patients to predict cardiotoxicity.
  3. There are many typos and grammar mistakes in all of manuscript. Please check and edit them.
  4. The citations are not incorrect: please check all citations
  • Line 84, no. 21 is incorrect
  • 20-23 were missing
  • Line 377, no. 60, 55, 56-58 are incorrect
  • Line 382, no 60-61 are incorrect
  • Lines 388-393, no. 67, 68, 69, 70-71 are incorrect

Minor comments

  1. 1: what are beta value and delta beta?
  2. 2: the lable is too small to read; no complete legends.
  3. 3: all words are too small to read; please explain TSS200 and TSS1500 in the legend.
  4. 4: why does only mRNA splcing have significance (FDR<0.05)?
  5. 5: all words are too small to read.
  6. 6: all words are too small to read.

Author Response

Reviewer 1

Major Comments:

  1. The authors should verify the results of DMR by (1) confirmed the differential methylation in patients by bisulfite sequencing or PCR-restriction enzyme digestion in RNA39, IRF6 and SLFN12 promoter region; (2) performing qPCR analysis to monitor their gene expression. These are important to demonstrate the relationship between DMR and cardiotoxicity. The reviewers make an excellent point. We have gone back and performed qPCR for the three genes. There was no expression of RNF39 but we obtained results for the other two. We have added to the methods about the analysis (page 5 line 183) and add the results (page 14 line 335)
  2. The aim 1 and aim 2 descriptions are confused. I suggest that just mention to identify the DMRs in patients to predict cardiotoxicity. The authors thank the reviewer for the comment and agree that stating both aims is confusing when most of the results are in regards to aim 2. We have removed mention of aim one to clarify the results.
  3. There are many typos and grammar mistakes in all of manuscript. Please check and edit them. We thank the reviewer for their identification of typo and errors. All reviewers have gone over the manuscript and corrected typos and grammatical mistakes.
  4. The citations are not incorrect: please check all citations. The reviewers are correct that the citations had become incorrect or missing. We have gone through and corrected all the citations and made sure they were all included.
  • Line 84, no. 21 is incorrect - fixed
  • 20-23 were missing- fixed
  • Line 377, no. 60, 55, 56-58 are incorrect- fixed
  • Line 382, no 60-61 are incorrect- fixed
  • Lines 388-393, no. 67, 68, 69, 70-71 are incorrect- fixed

Minor comments

  1. 1: what are beta value and delta beta? This has been explained in the caption and in the text (page 5 line 196)
  2. 2: the lable is too small to read; no complete legends. Text size has been increased and legend completed.
  3. 3: all words are too small to read; please explain TSS200 and TSS1500 in the legend. TSS200 and TSS1500 has been explained and text size increased.
  4. 4: why does only mRNA splcing have significance (FDR<0.05)? In this analysis those were the only pathways that were still significant after multiple correction but thought it still informative to see the other results as pathways that may be involved but just didn’t reach the level of significance
  5. 5: all words are too small to read. Text size increased.
  6. 6: all words are too small to read. Text size increased.

Reviewer 2 Report

The authors have presented the bioinformatics data effectively. This type of research will be helpful in personalized treatment approach for cancer patients. How much amount of blood was withdrawn from patients and is it possible for authors to include images of patient derived PBMC's? 

Author Response

Reviewer 2

The authors have presented the bioinformatics data effectively. This type of research will be helpful in personalized treatment approach for cancer patients. How much amount of blood was withdrawn from patients and is it possible for authors to include images of patient derived PBMC's? We thank the reviewer for their comments. 10ml of blood was collected before and after the first cycle of chemotherapy. We have updated the text (page 3 line 119). We do have an image of the PBMC but did not think it added to paper but we can add it to supplemental figures if the reviewer believes it is necessary.

Reviewer 3 Report

The paper by Bauer et al describes an interesting study that addresses the correlation between the methylation signature of peripheral blood mononuclear cells (PBMCs) and the risk of doxorubicin (DOX) cardiotoxicity in breast cancer patients, both prior to the start and after the first cycle of DOX-based chemotherapy. The final aim is to evaluate the gene methylation profile in PBMCs as a predictive marker of the risk of cardiotoxicity in breast cancer patients. The authors present the data suggesting that the extent of methylation at baseline indicates the potential to predict the subsequent development of cardiotoxicity. The paper is well-written and the findings are novel and very promising, but the study has important limitations and several problems that should be pointed out.  

MAJOR ISSUES:

  1. This study has several limitations such as a small number of patients examined and a big heterogeneity of patients in terms of the tumor type, age and race of patients. In fact, ER positive and triple negative patients were included in the study. The ER status may have a great impact on the analyzed data, but considering that only one ER-negative patient was in the test Abnormal LVEF group it is impossible to make any hypothesis in this direction. In my opinion the authors should exclude ER negative patients from the study and re-analyze the data only in ER positive group, and if possible, increase the number of ER+ patients.
  2. The experimental procedures are not clearly described and difficult to understand for non-experts in the Infinium Human Methylation 450 BeadChip approach and ChAMP R normalization method. The authors should describe the procedures in a more transparent way that can be understood by scientists not familiar with the applied approaches and provide a flow chart explaining the work flow of the study with a particular attention to the DMP and DMR methylation profile analysis.
  3. The methods of statistical analysis of the data should be clearly stated in a separate paragraph in Materials and Methods.

MINOR ISSUES:

  1. Simple summary: The authors should put more effort to summarize the paper in layman’s terms
  2. The Abstract must be shortened to 200 words (now more than 300 words)
  3. Many figures should be enlarged and in particular, the authors should increase the font size as it is absolutely unreadable in several figures. The font size of the text inside the figures should be comparable to the main text.
  4. Doxorubicin (DOX), should be introduced in simple abstract at the first mention “… DOX-based chemotherapy …”
  5. Line 14: “peripheral blood cells (PBCs)” should be “peripheral blood mononuclear cells (PBMCs)”
  6. Line 18 “peripheral blood cells” should be PBMC
  7. Line 20: the abbreviation LVEF should be explained
  8. All the above terms can appear as abbreviations in the Abstract
  9. “EF” abbreviation used in line 173 and in other 12 lines should be explained at the first mention.
  10. Table 1 legend is missing. Several abbreviations are not described.
  11. The LVEF data should be shown as the main figure

Author Response

Reviewer 3

MAJOR ISSUES:

  1. This study has several limitations such as a small number of patients examined and a big heterogeneity of patients in terms of the tumor type, age and race of patients. In fact, ER positive and triple negative patients were included in the study. The ER status may have a great impact on the analyzed data, but considering that only one ER-negative patient was in the test Abnormal LVEF group it is impossible to make any hypothesis in this direction. In my opinion the authors should exclude ER negative patients from the study and re-analyze the data only in ER positive group, and if possible, increase the number of ER+ patients. – We thank the reviewers for their suggestion but it would not be possible to increase the number of ER+ patients. With already few number of samples we are hesitant to remove the ER negative patient. Our results were still very significant and even the ER negative patient still clustered with patients with similar ejection fraction status. This seems to point that it is not the cancer but another factor that effects whether cardiotoxicity is a risk after DOX treatment.
  2. The experimental procedures are not clearly described and difficult to understand for non-experts in the Infinium Human Methylation 450 BeadChip approach and ChAMP R normalization method. The authors should describe the procedures in a more transparent way that can be understood by scientists not familiar with the applied approaches and provide a flow chart explaining the work flow of the study with a particular attention to the DMP and DMR methylation profile analysis. We thank the reviewers for there comment. We have rewritten the normalization method and added a citation to literature that explains the normalization method (page 4 line 149). We have also created a flow diagram to help explain the methylation analysis (Figure 1).
  3. The methods of statistical analysis of the data should be clearly stated in a separate paragraph in Materials and Methods. We thank the reviewers for their suggestion and have added 2.8. Statistical Analysis section (Page 5 line 191).

MINOR ISSUES:

  1. Simple summary: The authors should put more effort to summarize the paper in layman’s terms – Effort was taken to summarize the paper in layman’s terms (page 1 line 15)
  2. The Abstract must be shortened to 200 words (now more than 300 words) - The abstract has been reduced to less than 200 words (page 1 line 21).
  3. Many figures should be enlarged and in particular, the authors should increase the font size as it is absolutely unreadable in several figures. The font size of the text inside the figures should be comparable to the main text. All figures have been reproduced with larger font.
  4. Doxorubicin (DOX), should be introduced in simple abstract at the first mention “… DOX-based chemotherapy …” We have defined DOX in the simple abstract
  5. Line 14: “peripheral blood cells (PBCs)” should be “peripheral blood mononuclear cells (PBMCs)” – This has been corrected.
  6. Line 18 “peripheral blood cells” should be PBMC – This has been corrected.
  7. Line 20: the abbreviation LVEF should be explained – This has been corrected.
  8. All the above terms can appear as abbreviations in the Abstract – Reduced the words to less than 200 abstract may be more clear if we explain the abbreviations
  9. “EF” abbreviation used in line 173 and in other 12 lines should be explained at the first mention. – Added Ejection fraction (EF) status on first mention
  10. Table 1 legend is missing. Several abbreviations are not described. – Add description for abbreviations or did not abbreviate if only used once
  11. The LVEF data should be shown as the main figure - We thank the reviewer for the suggestion. We were not exactly sure which LVEF data should be the main figure. If its the one in the supplemental table we can move it to the main document if that would make the paper more clear.

Round 2

Reviewer 1 Report

Thanks for the authors' effort. They have significantly improved the manuscript. My only concern is Fig.8: please change the description of the y-axis to "Relative mRNA expression", and it is better if they can add error bars for the graphs.

Author Response

  1. Thanks for the authors' effort. They have significantly improved the manuscript. My only concern is Fig.8: please change the description of the y-axis to "Relative mRNA expression", and it is better if they can add error bars for the graphs. We thank the reviewer for their comments. We have changed the y-axis label per the suggestion on figure 8. We have also added error bars to the graphs. (page 14 line 305). An error was found in the original data and that was corrected and new graphs produced.

Reviewer 3 Report

The authors answered most issues addressed by the reviewer as well as added new data regarding the analysis of the gene expression of two differentially methylated genes IRF6, SLFN12 that showed inverse correlation with DNA methylation in patients with DOX cardiotoxicity. These new data improve the quality of the Ms, but they need to be presented  in a professional way.

Unfortunately, the authors were not able to increase the number of patients in the study and reduce their heterogeneity, which is the biggest limitation of this paper suggesting that the promising observations of this Ms are preliminary. Accordingly, the authors admit that a larger cohort of patients is needed to confirm the findings, but I think that Bauer et al should stress more that their observations are preliminary due to low number of patients analyzed. Since the data are statistically weak, the authors should point out to the well-done presentation of the methodological approach as suggested below. These and other minor issues should be addressed in the second revision.

MINOR ISSUES:

  1. The word “preliminary” should be used with regard to the overall findings of the paper. See above.
  2. In the minor point 11 “The LVEF data should be shown as the main figure” I meant that the data heart performance data LVEF from individual patients should be presented as the main figure in the paper. Now I realized that these data are actually missing as the data in Supplementary table 5 are not relative to LVEF measurements. The table 1 reports only the average group LVEF without the SD. These data should be implemented.
  3. It is not clear if the data in the figure 8 are statistically significant. The error bars are missing. It is also not clear how many patients’ mRNA samples were analyzed. All this information should be included in the legend. The original data regarding this experiment should be provided with Supplemental data.
  4. The flow chart inserted by the authors is quite helpful, but it does not show the PBMC and DNA isolation/preparation steps. Without these steps it is difficult to understand the method. The authors should add these steps to the flow chart.
  5. The legend below the flow chart should contain more relevant information, for example when looking at the chart the reader does not understand what normal and abnormal refers to. In addition, the squares indicating software packages should be distinguished from molecular biology methods, by marking the squares indicating the name of the software (f.e. black filling, white names). The point is that the flow chart should be self-explanatory.
  6. A brief description PBMCs isolation method should be supported by at least one reference with a full protocol.
  7. The bisulfite-modification method and DAMPK analysis should be supported by some references.
  8. In the Material and Methods section the authors repeat the full names of abbreviations previously explained in the abstract and the introduction section: Line 120: peripheral blood mononuclear cells (PBMCs); Line 124: left ventricle ejection fraction, which is unnecessary, but do not explain other abbreviations like: ChAMP, CpGAssoc and IDAT; Please, carefully check all the other acronyms.
  9. Line 17-18 sentence “The results showed that a significant differential methylation of PBMC …” should be clearer f.e.: “The results showed that significant differences in the DNA methylation pattern of PBMC were associated with the risk of cardiotoxicity.
  10. The authors should ask a native speaker English colleague to check the test as there are still some style problems.

Author Response

  1. The word “preliminary” should be used with regard to the overall findings of the paper. See above. We thank the reviewer’s for their comment. We have gone through the manuscript and made sure to include the word “preliminary” in regards to the overall findings. (page 1 line 19, page 1 line 33, page 3 line 103, page 17 line 327, page 18 line 427, page 19 line 446)
  2. In the minor point 11 “The LVEF data should be shown as the main figure” I meant that the data heart performance data LVEF from individual patients should be presented as the main figure in the paper. Now I realized that these data are actually missing as the data in Supplementary table 5 are not relative to LVEF measurements. The table 1 reports only the average group LVEF without the SD. These data should be implemented. We thank the reviewer for their clarification and suggestion. We have included the SD for the average LVEF percent. (page 6 line 234)
  3. It is not clear if the data in the figure 8 are statistically significant. The error bars are missing. It is also not clear how many patients’ mRNA samples were analyzed. All this information should be included in the legend. The original data regarding this experiment should be provided with Supplemental data. An error was found in the original data and that was corrected and new graphs produced. We have also added error bars to the graphs. (page 14 line 305) The original qPCR data has been added to supplemental data. (table 6-8).
  4. The flow chart inserted by the authors is quite helpful, but it does not show the PBMC and DNA isolation/preparation steps. Without these steps it is difficult to understand the method. The authors should add these steps to the flow chart. We have added the PBMC DNA isolation/prep steps to the flow chart. (page 4 line 141)
  5. The legend below the flow chart should contain more relevant information, for example when looking at the chart the reader does not understand what normal and abnormal refers to. In addition, the squares indicating software packages should be distinguished from molecular biology methods, by marking the squares indicating the name of the software (f.e. black filling, white names). The point is that the flow chart should be self-explanatory. We thank the reviewers for their comment. We have distinguished between software and biological methods. (page 4 line 141)
  6. A brief description PBMCs isolation method should be supported by at least one reference with a full protocol. We have added a brief description of the PBMCs isolation method with a reference. (page 3 line 124)
  7. The bisulfite-modification method and DAMPK analysis should be supported by some references. We have added a reference to the application note that describes this method. (page 5 line 145)
  8. In the Material and Methods section the authors repeat the full names of abbreviations previously explained in the abstract and the introduction section: Line 120: peripheral blood mononuclear cells (PBMCs); Line 124: left ventricle ejection fraction, which is unnecessary, but do not explain other abbreviations like: ChAMP, CpGAssoc and IDAT; Please, carefully check all the other acronyms. We thank the reviewers for identifying these redundancies. We have removed them. We have also explained IDAT in the text.  ChAMP and CpGAssoc are just the names of software tools. (page 5 line 158 & line  169)
  9. Line 17-18 sentence “The results showed that a significant differential methylation of PBMC …” should be clearer f.e.: “The results showed that significant differences in the DNA methylation pattern of PBMC were associated with the risk of cardiotoxicity. We thank the reviewer for their suggestion and have updated the text per their suggestion. (page 1 line 17)
  10. The authors should ask a native speaker English colleague to check the test as there are still some style problems. In response to the reviewers comment we have had the manuscript reviewed by an external colleague. The changes have been tracked in the word document.